# Investigation into Corrosive Wear of the CoCrFeNiTa$_x$ Laser-Clad Coatings on TC4 in the Neutral and Alkaline Circumstance

Yanan Yan, Jun Li *, Ruoliu Li, Meng Shao and Jing Li

School of Materials Science and Engineering, Shanghai University of Engineering Science, Shanghai 201620, China
* Correspondence: jacob_lijun@sues.edu.cn; Tel.: +86-21-67791198; Fax: +86-21-67791377

**Abstract:** CoCrFeNiTa$_x$ (x = 0, 1) coatings were prepared on the surface of TC4 by laser-cladding to improve the corrosive wear property of TC4 and extend its service life in corrosive media, ultimately aiming to improve its surface performance. The effects of Ta addition on the phase constituent and microstructure of the coatings were first investigated. Given the wide application of TC4 in corrosive media (even undergoing friction), the corrosive wear of TC4 covered with the coatings was especially focused in two media (neutral and alkaline). The results showed that the coatings were composed of primary α(Ti) and eutecticum (α(Ti) + Ti$_2$Ni) as the matrix and TiC as the reinforcement. The introduction of Ta increased the volume fraction of eutecticum and TiC and made the microstructure more uniform. The coating with Ta performed more outstanding corrosive wear resistance than the other two samples in the two media. The wear rate of the coating with Ta was 4.7458 × 10$^{-4}$ mm$^3$·N$^{-1}$·m$^{-1}$ in the neutral environment and 6.6808 × 10$^{-4}$ mm$^3$·N$^{-1}$·m$^{-1}$ in the alkaline environment, 64.45% and 61.79% lower than those of TC4, respectively. The wear mechanism of the samples is a combination of serious micro-cutting, active dissolution, and oxidation, and the introduction of Ta effectively improved the resistance to micro-cutting.

**Keywords:** laser cladding; TC4; multi-component alloy; microstructure; corrosive wear resistance; corrosion resistance

## 1. Introduction

Titanium alloys (especially Ti6Al4V) as the first practical titanium alloy have been successfully developed in America, and their usage amount has accounted for 75%–85% of the total amount of titanium alloys; they have been widely used in the automotive [1], marine [2], aerospace [3], and biomedical [4] industries due to the properties of low density, high yield strength, good corrosion resistance, and good biocompatibility [5–8]. However, their low hardness results in poor wear resistance (especially in sliding condition) [9], which greatly limits their practical application.

Many ceramic-particle-reinforced metal matrix composite coatings have been fabricated by laser-cladding on titanium alloys to improve their wear resistance [10,11]. Ceramic particle-reinforced metal matrix composite coatings could effectively advance the hardness of the TC4 surface, resulting in enhanced resistance to micro-cutting. However, wear resistance is not only dependent on the hardness of the coating but also on fracture toughness. High hardness is usually accompanied by the reduction in fracture toughness. Moreover, the rapid heating and cooling characteristic involved in laser cladding could cause large residual stress produced in the coating. The combination of the above two factors results in a high cracking susceptibility of the coating. The coating as a friction component undergoes alternating stress under long-term service, and the fatigue cracks initially and propagates easily, causing the damaged zones to suffer from brittle debonding. Therefore, improving

the fracture toughness as compensation for the reduction in hardness may endow the coating with more outstanding wear resistance.

Multi-component alloy coatings as an alternative are expected to solve the above shortcoming involved in the ceramic-particle-reinforced metal matrix coating. The atoms in a multi-component alloy are randomly arranged, resulting in a solid solution with a simple structure being easily formed and intermetallic compounds being greatly inhibited [12]. The formed solid solution with various elements presents large lattice distortion due to the difference among those, demonstrating increased strength and hardness. Moreover, the large lattice distortion reduces the rate of atomic diffusion in the alloy, thereby contributing to the formation of nanocrystalline or amorphous structure with excellent mechanical and electrochemical performance. Finally, the cocktail effect in multi-component alloy indicates that the fundamental properties of each element are incorporated into the overall properties of the alloy [13]. Therefore, the multi-component alloy often has excellent corrosion resistance and mechanical properties, such as high strength, outstanding wear resistance, and excellent creep resistance [14,15]. Some multi-component coatings have been explored and fabricated on titanium alloys by laser cladding [16–19]. At present, studies on wear resistance of laser-clad multi-component alloy coatings on TC4 mainly focused on the wear behavior in air at room or high temperature. TC4 as a corrosion-resistant material is widely applied in the engineering field, including bolts, pads, riveting, and connecting pipes, which are often used in corrosive environment [20–22], such as seawater, oilfield, and potassium formate completion fluid. Its outstanding corrosion resistance originates from a layer of dense and inert oxidation film with a thickness of approximately 1–10 nm, which is spontaneously formed and tightly adheres to its surface. However, its strong protection role is only demonstrated in static corrosive environment. Some TC4 components often contact with other components and perform the relative motion, thus suffering from strong effects from small hard solid particles flowing at high speed in the corrosion medium. The oxidation film easily suffers from destruction, causing the chemical dissolution of the exposed substrate. Moreover, the exposed substrate is easily cut under the mechanical action due to its low hardness. The material loss in a corrosion-wear environment is much higher than that in single-wear or single-corrosion environment. Therefore, the results of wear resistance in air could not be used to predict the actual service performance in corrosive environment. Current investigations have proven that the interactions between mechanical action and corrosion media could accelerate the failure of the friction components under corrosive circumstance. A dense and stable passive film formed on the coating in corrosive solution could be destroyed during the dry sliding process, resulting in the coating being re-exposed to the solution and suffering from active dissolution. Meanwhile, the active dissolution could cause the formation of corrosion pits similar to the notches due to the inhomogeneous compositions at different zones. The initiation and propagation of the cracks are promoted by the stress concentration at the notch tip, leading to the acceleration of material loss. Therefore, investigating the wear behavior of multi-component alloy coatings fabricated on TC4 is very indispensable.

In the present study, the CoCrFeNi alloy system, as a common and promising alloy, was chosen to be the cladding material [23,24]. Ta was introduced into the system to further improve the tribocorrosion performance. Ta, a refractory metal ($T_m$ = 2996 °C) [25], presents outstanding corrosion resistance in various acids, salt solutions, and organic chemicals due to a stable passive oxide film ($Ta_2O_5$) being rapidly formed in the corrosion circumstance. Moreover, the large difference in radius between Ta and the other elements in CoCrFeNi alloy could enhance the solid solution strengthening effect, and may be favorable to the formation of secondary phase [26], which is conducive to the improvement in wear resistance. Lv et al. [27] prepared laser-clad composite coatings on the surface of TC4 by using NiCrBSi alloy powder with different contents of TaC and investigated the wear resistance at high temperature of coatings. With the increase in TaC content, the wear volume of the coatings showed an opposite trend, indicating the improvement of wear resistance. When the content of TaC was 40 wt.%, the wear volume was only

$2.49 \times 10^{-7}$ mm$^3$, which was a decrease by 97.8% compared with that of the substrate $(1.15 \times 10^{-5}$ mm$^3)$, demonstrating the best wear resistance. Hu et al. [28] fabricated TiB, TiB$_2$, and TiC reinforced Ti$_2$Ni/TiNi matrix coatings on TC4 via laser cladding. Different contents of TaC were also added into the coating material. The results showed that the addition of TaC greatly improved the resistance to corrosion and pitting corrosion in 0.1 mol/L HCl solution, and the improvement effect increased with the increase in TaC content. When the content of TaC was up to 20 wt.%, the E$_{corr}$ of the coating achieved the maximum value $-0.086$ V ($-0.423$ V for the substrate), and the i$_p$ was down to the minimum value of $2.1763 \times 10^{-7}$ A·cm$^{-2}$ ($1.0314 \times 10^{-7}$ A·cm$^{-2}$ for the substrate).

Thus, in this paper, the CoCrFeNiTa$_x$ (x = 0, 1) alloy system was selected to be fabricated on TC4 via laser-cladding. The effect of Ta addition on the phase composition and microstructure was studied. Tribocorrosion performance was especially highlighted, which was evaluated by the pin-disk reciprocating sliding mode in neutral (3.5 wt.% NaCl solution, pH = 7) and alkaline (NaOH solution, pH = 11) environments. The wear mechanism was also revealed.

## 2. Materials and Methods

### 2.1. Preparation of the Coatings

In this experiment, TC4 alloys applied as the substrates of laser cladding were cut into sheets by a DK7730C EDM wire-cutting machine (Haishu Guoding Numerical Control Machinery Co., Ltd., Ningbo, China), and the diameter and thickness of the sheets were 50 and 10 mm, respectively. The sheets contained a large amount of α phases and traces of β phases after being subjected to annealing. The annealing treatment aimed to eliminate the α′ phase that precipitated during the solution treatment, as it could produce unfavorable effects to the comprehensive performance. The samples were ground with 240# SiC sandpaper to remove the oxidation film on the substrate surface. Then, the samples were cleaned in acetone with an ultrasonic cleaner for 30 min. Commercial cladding powders, including Co, Cr, Fe, Ni, and Ta (≥99.5 wt.% in purity and 53–105 µm in size), with two different mole ratios (CoCrFeNiTa$_x$, x = 0, 1) were prepared. The powders were dried at 80 °C for 10 h and then placed into a Teflon grinding jar with agate balls (eight with a diameter of 6 mm, six with a diameter of 10 mm and four with a diameter of 15 mm) in it. Finally, the jar was placed on a GQM-2-15 grinding miller at 300 r·min$^{-1}$ for 10 h to mix the powders uniformly. Agate balls were introduced into the grinding jar to grind some coarse particles into fine particles, thus improving the mixing uniformity.

The substrates were placed into a circular model measuring 50.2 mm in inner diameter and 11.0 mm in height, and 4 wt.% polyvinyl alcohol was applied to the surface of substrates. The mixed powders were uniformly spread on the substrate and then compressed for 120 s under a load of 30 MPa by a BJ-30 tablet machine (Tianjin Bojun Technology Co., Ltd., Tianjin, China) to obtain a pre-prepared coating with 0.8 mm in thickness. Before the laser-cladding was carried out, the samples were dried at 80 °C for 1 h.

Laser-cladding was carried out by a YSL-5000 fiber laser system, and the parameters of the process were as follows: 3 kW for power, 5 mm·s$^{-1}$ for scanning rate, and 6 mm for spot diameter.

### 2.2. Microstructural Characterization

The phase compositions of the coatings were characterized by a PANalytical X'Pert Pro X-ray diffractometer (Malvern Panalytical Co., Ltd., Shanghai, China) with Cu Kα radiation (λ = 0.1540560 nm). The microstructural morphologies and chemical compositions of the cross-sections for coating without and with Ta were observed by scanning electron microscopy (SEM, Hitachi S-3400, Hitachi, Tokyo, Japan) coupled with energy-dispersive spectrometry (EDS, GENESIS EDAX, EDAX Inc., Philadelphia, PA, USA). Before the observation was performed, metallographic samples were prepared by the following steps corresponding to wire-cutting: grinding with sandpapers from 360# to 2000#; polishing

with 2.5 μm diamond grinding paste; and etching for 40 s in a mixture containing 4 mL deionized water, 6 mL $HNO_3$ and eight drops of HF.

### 2.3. Electrochemical Performance

The electrochemical performance of the samples was examined on a CHI 760E electrochemical workstation in neutral (3.5 wt.% NaCl solution, pH = 7) and alkaline (NaOH solution, pH = 11) environments at room temperature. The classical three-electrode system was selected for measurement, in which the samples were the working electrode (diameter, 2.5 mm), a platinum plate was the auxiliary electrode, and a saturated calomel electrode was the reference electrode. In particular, the saturated calomel electrode was corroded when the test environment was a NaOH solution. Thus, a salt bridge must be used for the test. Before the tests were conducted, the samples were left in the solution to stand for 30 min. Tafel polarization curves were plotted under a potential window ranging from −0.8 V to 1.0 V with a scanning speed of $1 \text{ mV·s}^{-1}$. The surfaces of the samples after the electrochemical test were observed to analyze the chemical compositions and their valences by X-ray photoelectron spectroscopy (XPS, ESCALAB 250XI, Thermo Fisher Scientific, Waltham, MA, USA).

### 2.4. Friction and Wear in Corrosive Media

A CFT-1 functional friction tester (Lanzhou Zhongke Kaihua Technology Development Co., Ltd., Lanzhou, China) was used to evaluate wear resistance, with a load of 20 N, a sliding time of 180 min, a rotating speed of 600 rpm, and a reciprocating distance of 3 mm. A hard YG6 alloy ball (94 wt.% WC and 6 wt.% Co; 5 mm in diameter) with HRA 91.5 in hardness was selected as the counterpart and replaced by a new one after each test. The tests were performed in neutral (3.5 wt.% NaCl solution, pH = 7) and alkaline (NaOH solution, pH = 11) environments, separately. The friction coefficients of the substrate and the coatings were recorded in real time during the friction process. After sliding wear testing was performed, the wear volumes of the samples were measured by a surface mapping profile meter. The wear morphologies and distribution of the elements of the samples undergoing wear tests were observed by SEM and EDS. The wear rates (K) of the samples were also obtained using Archard's equation as follows:

$$K = \frac{V}{N \times d} \tag{1}$$

where *V* represents the wear volume ($mm^3$), N indicates the applied load (N), and d denotes the sliding distance (m).

## 3. Results and Discussion

### 3.1. XRD Analyses

Figure 1 demonstrates the XRD patterns of the coatings without and with Ta. The patterns of the coatings are similar in the number and position in the diffraction peak, implying that the phase constituents in the two coatings are also similar. Eight peaks could be clearly detected at 2θ = 39.115°, 2θ = 40.095°, 2θ = 41.607°, 2θ = 45.393°, 2θ = 57.977°, 2θ = 70.606°, 2θ = 72.750°, and 2θ = 86.435°, among which the strongest peak appeared at 40.095°. By comparing the d values of the abovementioned peaks with those in JCPDS cards, the coatings could be confirmed to be mainly composed of α(Ti) (No. 44-1294), $Ti_2Ni$ (No. 18-0898) and a small quantity of TiC (No. 32-1383). Table 1 shows the d value of the detected peaks and those related to the corresponding phases in JCPDS cards. It can be seen that the maximum differences of d were 0.0118 for α(Ti), 0.0101 for $Ti_2Ni$, and 0.0151 for TiC, indicating that the calibration results are convincing. A clear inspection revealed that all the diffraction peaks moved to the left in the coating with Ta compared with that without Ta, implying that the introduction of Ta exacerbated the lattice distortion because the atomic radius of Ta is larger than those of Co, Cr, Fe, Ni, Ti, and Al involved in the coatings (Ta: 1.48 Å; Co: 1.26 Å; Cr: 1.27 Å; Fe: 1.27 Å; Ni: 1.24 Å; Ti: 1.45 Å; and Al: 1.43 Å).

The severe lattice distortion could produce a positive effect on solid solution strengthening, which could contribute to the improvement in hardness and resistance to micro-cutting of the coating when in contact with the other component and performing the relative motions.

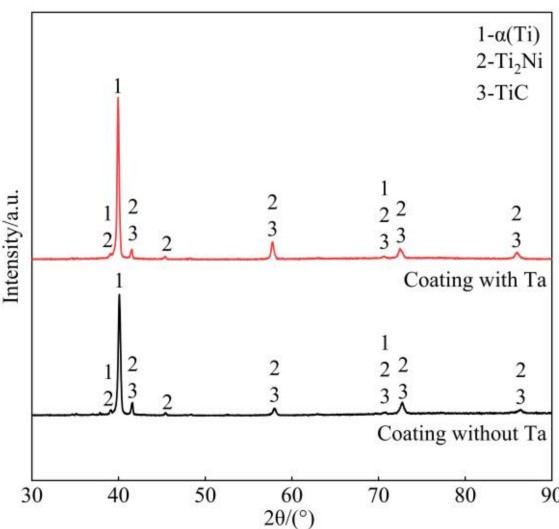

**Figure 1.** X-ray diffraction patterns of the coatings.

**Table 1.** XRD data for coating without and with Ta.

| Experiment Data d (nm) | | $\alpha$(Ti) No. 44-1294 d (nm) | Ti$_2$Ni No. 18-0898 d (nm) | TiC No. 32-1383 d (nm) |
|---|---|---|---|---|
| Coating without Ta | Coating with Ta | | | |
| 2.3011 | 2.3024 | 2.3070 | 2.3020 | - |
| 2.2471 | 2.2548 | 2.2430 | - | - |
| 2.1688 | 2.1727 | - | 2.1710 | 2.1637 |
| 1.9963 | 1.9954 | - | 1.9940 | - |
| 1.5894 | 1.5951 | - | 1.5850 | 1.5800 |
| 1.3329 | 1.3349 | 1.3320 | 1.3340 | 1.3400 |
| 1.2988 | 1.3032 | - | 1.2940 | 1.3047 |
| 1.1249 | 1.1294 | - | 1.1330 | 1.1200 |

### 3.2. Microstructural Characterization

The cross-sectional macro-morphology of the coatings is illustrated in Figure 2. A smooth wave-like interface divides the cross section into the coating and the substrate, indicating that a strong metallurgical bonding was formed between the two. The maximum thickness and width of the coatings are approximately 2.74/6.46 and 2.82/6.68 mm for the coatings without and with Ta, respectively. The profile of the coatings presented a crescent shape, that is, the thickness is largest in the central zone and gradually decreased along the sides. This finding should be associated with the Gaussian distribution in laser energy. Careful examination showed that the coatings are very dense, showing typical melting and solidification characteristics. Moreover, no cracks, pores, nor inclusions were observed.

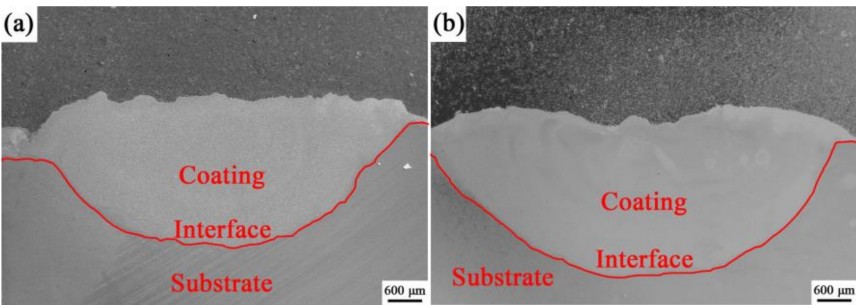

**Figure 2.** The cross-sectional macro-morphology of coatings: (**a**) without Ta, (**b**) with Ta.

　　　　Figure 3 indicates the microstructure of the two coatings under different magnifications. For the coating without Ta, a large number of irregular grains were uniformly distributed in the coating, among which wide and irregular honeycomb-like grain boundaries were connected with each other (Figure 3a). When Ta was introduced into the coating, the irregular grains were transformed into regular equiaxed grains with comparatively smooth edges, accompanied with honeycomb-like grain boundary that was greatly reduced in volume fraction (Figure 3b). A clear observation revealed that some fine dendrites were wrapped in the grains of the two coatings, and their volume fraction increased upon introducing Ta. Figure 3c–f show the images with high magnification. Besides the grains (marked as 1 and 5) and dendrites (marked as 4 and 8), the other two morphological phases could be obviously observed in the grain boundary, corresponding to the raised zones (marked as 3 and 7) and shrunken zones (marked as 2 and 5). EDS was applied to identify the chemical compositions of the abovementioned phases, and the results are shown in Table 2. Zone 1 is rich in Ti (67.04 at.%). Small quantities of Al (9.81 at.%), V (3.51 at.%), Cr (5.18 at.%), Fe (4.25 at.%), Co (4.16 at.%), and Ni (4.49 at.%) were also detected. Besides those elements, Ta was found in Zone 5. Combined with the XRD results, the grains in the two coatings could be confirmed as a $\alpha$(Ti) solid solution, in which $\alpha$-Ti with close-packed hexagonal structure played the role of solvent, and other alloying elements acted as solute. Zone 4 is rich in Ti and C, indicating the dendrites could be determined as TiC secondary solid solution. When Ta was introduced in the coating, Ta was involved in TiC, which displaced Ti atoms in the lattice due to their similar atomic radius. The raised zones could be confirmed as $\alpha$(Ti) due to their similar chemical compositions with the grains. The shrunken zone (Zone 2) in the coating without Ta mainly contained Ti (55.28 at.%) and Ni (11.27 at.%), followed by Al (5.19 at.%), Cr (3.69 at.%), Fe (9.18 at.%), and Co (12.25 at.%). The zone could be identified as $Ti_2Ni$, a secondary solid solution. As mentioned above, the atomic radius values of Ti, Al, and Cr are similar, and the electronegativity values among the three elements are about the same (Ti: 1.54, Al: 1.61, and Cr: 1.66), suggesting that the Al and Cr atoms could replace the Ti atoms in the $Ti_2Ni$ lattice. In the same manner, the Ni atoms in $Ti_2Ni$ could be substituted for Co and Fe atoms due to the close values in atomic radius and electronegativity (Ni: 1.24 Å in atomic radius, 1.91 in electronegativity; Co: 1.26 Å in atomic radius, 1.88 in electronegativity; and Fe: 1.27 Å in atomic radius, 1.83 in electronegativity). In addition, the atomic ratio of Ti + Al + Cr to Ni + Co + Fe was calculated as 1.96: 1, relatively close to 2:1. Similarly, the shrunken zone (Zone 6) in the coating with Ta was $Ti_2Ni$. Based on the above analyses, the microstructure consisted of primary $\alpha$(Ti), eutecticum ($\alpha$(Ti) + $Ti_2Ni$), and reinforcement TiC. The increase in the number of TiC and eutecticum is closely associated with Ta.

**Table 2.** Chemical compositions of marked zones shown in Figure 3.

| Coating | Zones | Elements (at.%) | | | | | | | | |
|---|---|---|---|---|---|---|---|---|---|---|
| | | C | Al | Ti | V | Cr | Fe | Co | Ni | Ta |
| Coating without Ta | 1 | 1.56 | 9.81 | 67.04 | 3.51 | 5.18 | 4.25 | 4.16 | 4.49 | - |
| | 2 | 1.20 | 5.19 | 55.28 | 1.93 | 3.69 | 9.18 | 12.25 | 11.27 | - |
| | 3 | 1.26 | 9.87 | 67.26 | 3.13 | 5.62 | 4.66 | 4.21 | 4.00 | - |
| | 4 | 1.50 | 4.31 | 90.12 | 0.38 | 0.95 | 0.81 | 0.78 | 1.15 | - |
| Coating with Ta | 5 | 2.29 | 8.07 | 60.76 | 3.01 | 5.12 | 3.80 | 2.82 | 1.85 | 12.27 |
| | 6 | 1.32 | 6.19 | 58.02 | 2.00 | 3.15 | 6.00 | 7.33 | 11.30 | 4.69 |
| | 7 | 2.04 | 7.91 | 61.55 | 3.42 | 4.26 | 4.02 | 2.78 | 1.67 | 12.35 |
| | 8 | 1.44 | 4.40 | 78.87 | 0.96 | 0.83 | 1.36 | 1.33 | 1.34 | 9.47 |

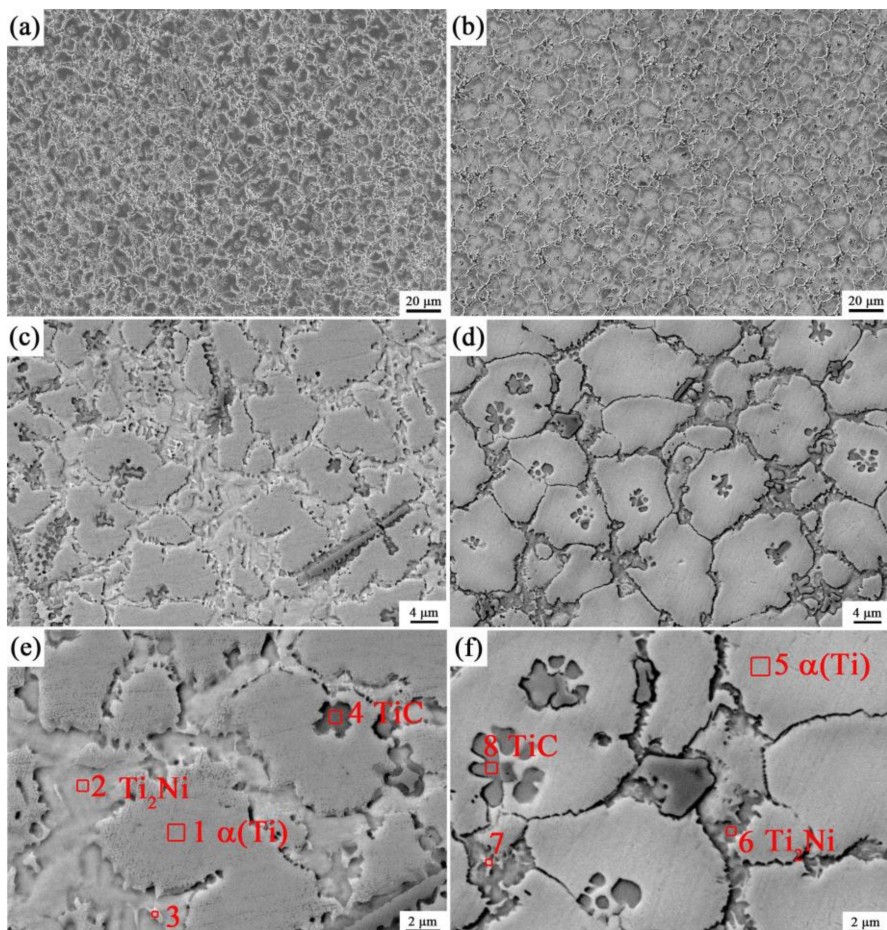

**Figure 3.** Microstructure of the coatings: (**a**) SE (secondary electron) image of coating without Ta; (**b**) SE image of coating with Ta; (**c**,**e**) BSE (back-scattered electrons) image of coating without Ta; (**d**,**f**) BSE image of coating with Ta.

### 3.3. Corrosion Performance

Figure 4 indicates the potentiodynamic anodic polarization curves of the substrate and the coatings in neutral environment (3.5 wt.% NaCl solution, pH = 7), and the curves of the samples presented a similar shape. Some characteristic parameters (Table 3) in terms of corrosion potential ($E_{corr}$), corrosion current density ($i_{corr}$), and current density ($i_s$) in the comparatively stable corrosion zone were used to evaluate the electrochemical performance of the substrate and the coatings.

**Table 3.** Electrochemical parameters obtained from Figure 4.

| Sample | $E_{corr}$/V | $i_{corr}$/A·cm$^{-2}$ | $i_s$/A·cm$^{-2}$ |
|---|---|---|---|
| Substrate | −0.359 | $2.591 \times 10^{-6}$ | $3.535 \times 10^{-6}$ |
| Coating without Ta | −0.367 | $1.412 \times 10^{-6}$ | $1.974 \times 10^{-6}$ |
| Coating with Ta | −0.360 | $1.582 \times 10^{-7}$ | $3.507 \times 10^{-7}$ |

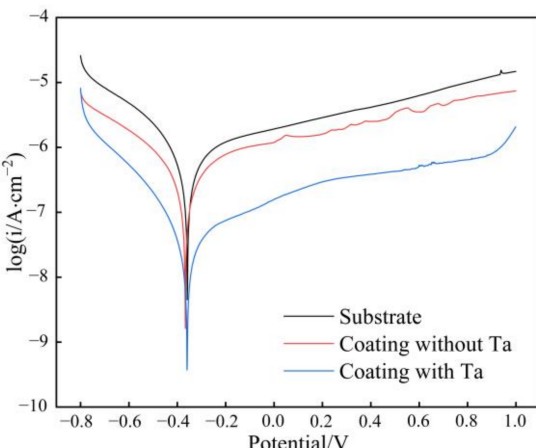

**Figure 4.** Potentiodynamic anodic polarization curves of the substrate and the coatings in the neutral environment (3.5 wt.% NaCl solution).

$E_{corr}$ is defined as the potential at which the material reaches a stable corrosion state. This parameter characterizes the ability of the electrode to lose electrons, and it could be applied to identify the corrosion tendency of a material. When a material is in contact with another material in a corrosion system, the material with a more negative $E_{corr}$ is inclined to act as the anode and suffers from serious corrosion by losing electrons. According to the results in Table 3, the $E_{corr}$ values of the substrate and coatings are similar ($-0.359$ V for the substrate, $-0.367$ V for the coating without Ta, and $-0.360$ V for the coating with Ta). The $i_{corr}$ is the corrosion current density corresponding to $E_{corr}$, which represents the corrosion rate of the material. The $i_{corr}$ value of the coating without Ta is $1.412 \times 10^{-6}$ A·cm$^{-2}$, reduced by 45.5% compared with that of the substrate ($2.591 \times 10^{-6}$ A·cm$^{-2}$). The $i_{corr}$ value of the coating with Ta was further reduced to $1.582 \times 10^{-7}$ A·cm$^{-2}$, which is an order-of-magnitude lower than that of the coating without Ta. This finding indicated that laser cladding could effectively improve the corrosion resistance of the substrate by fabricating the two coatings on it, and the introduction of Ta further strengthened the corrosion resistance. When the applied potential exceeded $E_{corr}$, the electrode entered into an active state. The current sharply increased along with the increase in potential. In this stage, metallic elements were subject to be oxidized, and they entered into the electrolyte in the form of ions. When the applied potential was further enhanced, the current density increased slowly and gradually tended to be a stable value ($i_s$) due to the formation of a uniform and dense passive film on the sample surface, indicating that the electrode already entered into a comparatively stable corrosion state. The corrosion rate in the comparatively stable corrosion state could be described by $i_s$, and the results demonstrated the following order: substrate ($3.535 \times 10^{-6}$ A·cm$^{-2}$) > coating without Ta ($1.974 \times 10^{-6}$ A·cm$^{-2}$) > coating with Ta ($3.507 \times 10^{-7}$ A·cm$^{-2}$). Evidently, the corrosion rate of the coating with Ta in the comparatively stable corrosion state is lower than that of other samples, suggesting that the oxidation film formed on the coating with Ta could effectively protect the sample from corrosion.

The above analyses confirmed that laser cladding the coating of CoCrFeNiTa could improve the corrosion resistance of the substrate in neutral environment. XPS was applied to detect the compositions and their corresponding valences of the oxidation film formed on the coating with Ta in 3.5 wt.% NaCl solution to further investigate the relationship between corrosion resistance and passive film (Figure 5). Figure 5a shows the survey spectrum of the oxidation film formed on the surface of coating with Ta, revealing that the oxidation film was composed of Ta, Co, Cr, Fe, Ni, Ti, and O. This finding indicated that the alloy was subject to oxidization. The chemical valences of the metallic elements could be obtained from the high-resolution narrow spectra (Figure 5b–g). The spectrum of Ta$_{4f}$ could be fitted into three peaks at 26.00, 27.9, and 22.00 eV in Figure 5b, indicating that Ta existed in the form of Ta$_2$O$_5$. For Co, two strong peaks were observed at 795.00 and 779.90 eV (Figure 5c),

corresponding to the $Co_{2p1/2}$ and $Co_{2p3/2}$ peaks, respectively, proving the presence of CoO. Figure 5d illustrates the $Cr_{2p}$ peak, which was composed of the $Cr_{2p1/2}$ peak at 586.4 eV and the $Cr_{2p3/2}$ peak at 576.8 eV. The two peaks confirmed the formation of $Cr_2O_3$ during the electrochemical tests. Two strong peaks of $Fe_{2p}$ could be clearly detected at 710.2 and 725.5 eV in Figure 5e, confirming the existence of $Fe_3O_4$. In Figure 5f, the raw line was not smooth, but two peaks could still be found at 854.5 and 871.9 eV, indicating the synthesis of NiO. Two strong peaks at 458.5 and 464.19 eV were also observed in the $Ti_{2p}$ spectrum in Figure 5g, which fit the standard peak of Ti in $TiO_2$ well. Therefore, the oxidation film formed on the surface of coating CoCrFeNiTa was composed of $Ta_2O_5$, CoO, $Cr_2O_3$, $Fe_3O_4$, NiO, and $TiO_2$. The above analyses showed that the following reactions occurred during the electrochemical test:

$$4Ta + 5O_2 = 2Ta_2O_5 \tag{2}$$

$$2Co + O_2 = 2CoO \tag{3}$$

$$4Cr + 3O_2 = 2Cr_2O_3 \tag{4}$$

$$3Fe + 2O_2 = Fe_3O_4 \tag{5}$$

$$2Ni + O_2 = 2NiO \tag{6}$$

$$Ti + O_2 = TiO_2 \tag{7}$$

The change in the standard Gibbs free energy ($\Delta G^\theta$) for the preceding reactions at room temperature (298 K) could be calculated as follows: $-3822.29$ KJ·mol$^{-1}$ for Reaction (2), $-786.50$ KJ·mol$^{-1}$ for Reaction (3), $-2096.27$ KJ·mol$^{-1}$ for Reaction (4), $-1015.44$ KJ·mol$^{-1}$ for Reaction (5), $-424.95$ KJ·mol$^{-1}$ for Reaction (6), and $-889.53$ KJ·mol$^{-1}$ for Reaction (7). Obviously, the $\Delta G^\theta$ values of these reactions are all negative, so all these reactions could occur spontaneously at room temperature. In thermodynamics, the smaller the $\Delta G^\theta$ value is, the easier it is for the reaction to occur, so the order of these reactions is Reaction (2) > Reaction (4) > Reaction (5) > Reaction (7) > Reaction (3) > Reaction (6). The corresponding oxides formed is in the order of $Ta_2O_5$ > $Cr_2O_3$ > $Fe_3O_4$ > $TiO_2$ > CoO > NiO. Thus, compared with the substrate rich in Ti, the Cr and Fe introduced to the cladding material could promote the formation of oxidation film, and the addition of Ta could further accelerate this process. For the coating, the compactness of the oxidation film formed on the surface of the coating played the essential role in the corrosion resistance of the coating. In the process of electrochemical test, metallic elements were oxidized into oxides and formed a thin oxidation film on the electrode surface, during which the oxidized zone could suffer from expansion or shrinkage and result in the creation of internal stress within it. The Pilling–Bedworth ratio (PBR) was used to reflect the stress in the oxidation film, and it is defined as the change in volume of a given metal element (X) before and after the oxidation ($AX + BO_2 = CX_mO_n$), which is also applicable in aqueous solutions [29]. PBR could be calculated using the following formula:

$$PBR = \frac{C\rho_X Z_{X_mO_n}}{A\rho_{X_mO_n} Z_X} \tag{8}$$

where *A* and *C* denote the mole number of the metal element (X) and the corresponding oxide ($X_mO_n$), respectively; $\rho_X$ and $\rho_{X_mO_n}$ refers to the density of X and $X_mO_n$, respectively; $Z_{X_mO_n}$ and $Z_X$ signify the atomic weight of X and the molecular weight of $X_mO_n$; and *X* and $X_mO_n$ represent the reactant and the product.

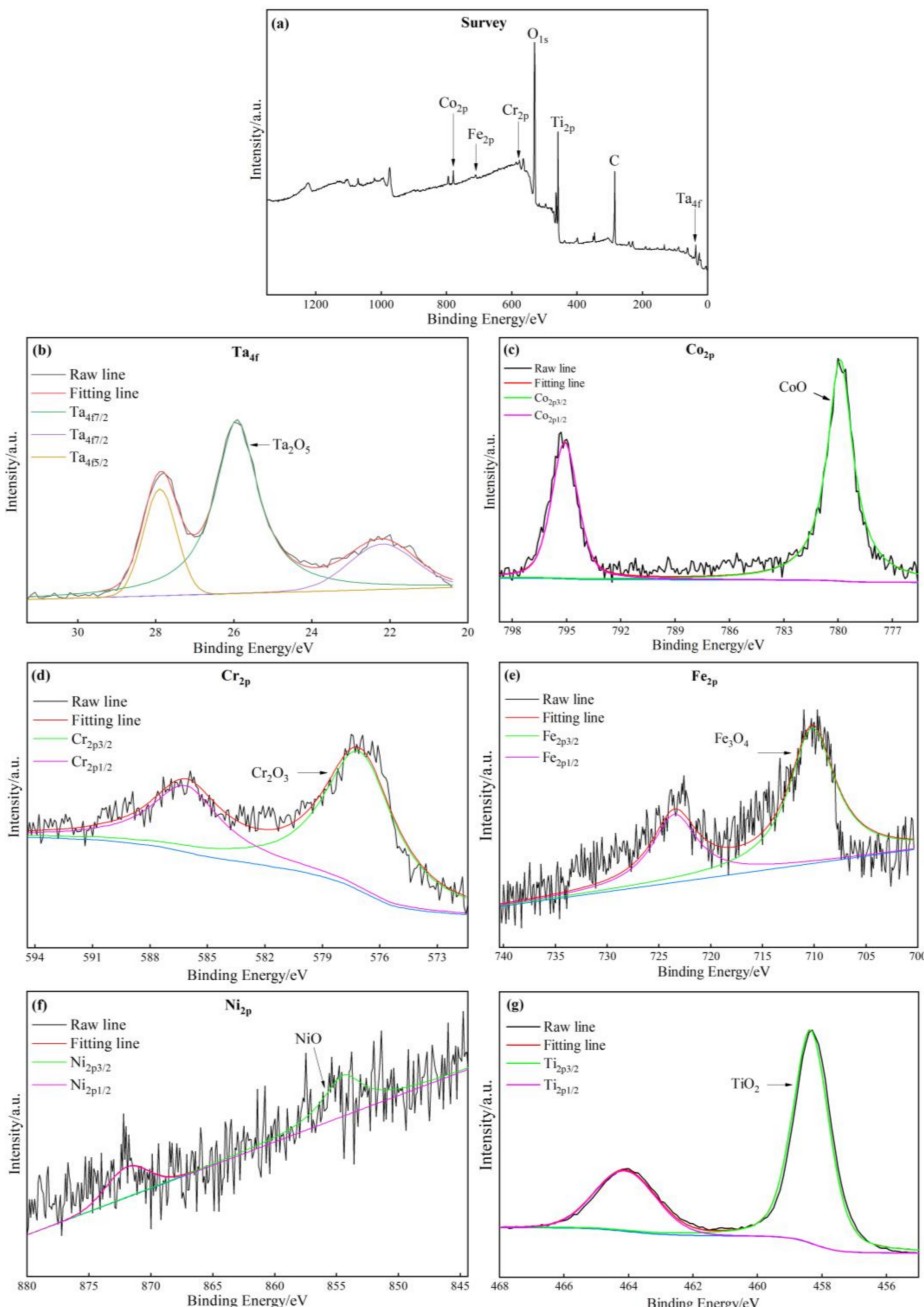

**Figure 5.** XPS spectra of the oxide film formed on the surface of coating with Ta in 3.5 wt.% NaCl solution: (**a**) survey; (**b**) $Ti_{2p}$; (**c**) $Co_{2p}$; (**d**) $Cr_{2p}$; (**e**) $Fe_{2p}$; (**f**) $Ni_{2p}$; and (**g**) $Ta_{4f}$.

When PBR < 1, the corrosion resistance of the oxidation film is poor due to the existence of tension stress, which results in incomplete coverage of the oxidation film on the sample surface, accompanied with a porous structure. When PBR > 1, the compressive stress exists in the oxidation film, and the formed oxide film is compact and complete, which could effectively protect the sample from corrosion. With the increase in PBR, the volume expansion of the sample is increased, and the oxidation film is denser. However, an excessive PBR value is not conducive to the improvement in the corrosion resistance of the sample due to the increase in cracking and debonding susceptibility of the oxidation film caused by the excessive stress. For Reactions (2)–(7), the values of PBR were calculated as 2.47, 1.76, 2.02, 2.10, 1.70, and 1.77, respectively. Compared with the substrate, the addition of Fe and Cr into the coating could effectively improve the compactness of the oxidation film because of the higher PBR value than that of Ti. However, the PBR of $Ta_2O_5$ is 2.47,

which revealed that the oxidation film has a tendency to rupture. Meanwhile, the PBR of Co and Ni is between 1 and 2, in which the oxidation film could not only protect the coating effectively with a density structure, but also reduce the stress of the oxidation film, resulting in the tight binding between the oxidation film and the sample. This finding is the reason for the improvement in the corrosion resistance of the coating after the addition of Ta.

Electrochemical testing was also carried out to investigate the corrosion resistance of the samples in alkaline environment (NaOH solution, pH = 11), and the corresponding Tafel curves are shown in Figure 6. The characteristic parameters obtained from the curves are listed in Table 4. A new parameter $E_{s-a}$ is obtained in Table 4, and it is denoted as the breakdown potential of the passive film. When the applied current reaches $E_{s-a}$, the passive film is destroyed, and the electrode enters the re-active state.

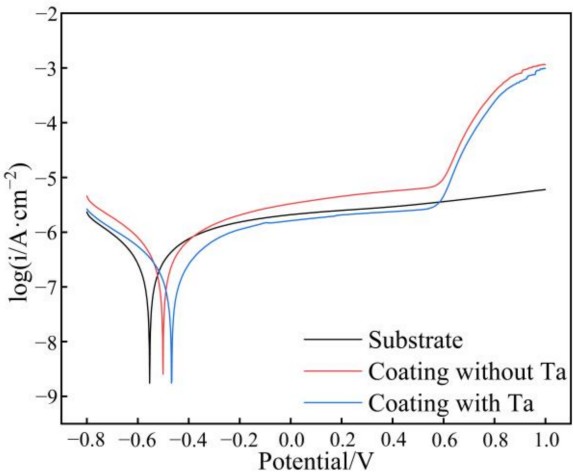

**Figure 6.** Potentiodynamic anodic polarization curves of the samples in NaOH solution (pH = 11).

**Table 4.** Electrochemical parameters obtained from Figure 6.

| Sample | $E_{corr}$/V | $E_{s-a}$/V | $i_{corr}$/A·cm$^{-2}$ | $i_s$/A·cm$^{-2}$ |
|---|---|---|---|---|
| Substrate | −0.554 | - | $1.182 \times 10^{-6}$ | $2.629 \times 10^{-6}$ |
| Coating without Ta | −0.501 | 0.588 | $1.554 \times 10^{-6}$ | $4.022 \times 10^{-6}$ |
| Coating with Ta | −0.468 | 0.568 | $7.700 \times 10^{-7}$ | $2.079 \times 10^{-6}$ |

As shown in Table 4, the values of $E_{corr}$ of the substrate and the coatings showed a gradual upward trend (−0.554 V for substrate, −0.501 V for coating without Ta, and −0.468 V for coating with Ta). This result showed that the corrosion tendency of the substrate was reduced by laser-cladding CoCrFeNi, and the effect was more obvious after introducing Ta. With regard to $i_{corr}$, the result showed a downward tendency as follows: $1.554 \times 10^{-6}$ A·cm$^{-2}$ (coating without Ta), $1.182 \times 10^{-6}$ A·cm$^{-2}$ (substrate), and $7.70 \times 10^{-7}$ A·cm$^{-2}$ (coating with Ta). Although the $i_{corr}$ value of the coating without Ta was a little higher than that of the substrate, the value of coating with Ta decreased by 34.86%. The $i_s$ value showed an upward trend with the following order: coating with Ta ($2.079 \times 10^{-6}$ A·cm$^{-2}$), substrate ($2.629 \times 10^{-6}$ A·cm$^{-2}$), and coating without Ta ($4.022 \times 10^{-6}$ A·cm$^{-2}$). Evidently, the corrosion rate of the coating with Ta in the comparatively stable corrosion state was lower than that of the other samples, suggesting that the oxidation film formed on the coating with Ta could protect the sample from corrosion effectively.

The passive film formed on the surface of the coatings obviously broke down with the increase in potential, which is relatively related to the strong corrosive of the NaOH solution. The passive film was composed of various oxides, and micro-gaps exist between oxides. The NaOH solution could penetrate into the passivation film from the gaps. Moreover,

the anodic Tafel slope of the coatings was significantly higher than the cathodic Tafel slope, indicating that the kinetic of corrosion of the coatings in NaOH solution is mainly controlled by the anode [30]. The anodic dissolution process in electrochemical testing is the entry of metal as ions into the corrosion solution. According to the study on the microstructure of the coating, the coating is a non-single phase coating, and the potential of different phases differs. After the NaOH solution entered the passivation film, different phases formed micro-galvanic cells [31]. The passivation film and matrix could also form micro-cells because of the same principle, thus accelerating the anodic dissolution, which leads to the appearance of the cavitation in the passivation film. The vacuoles gradually accumulate into a complete crack, finally leading to the cracking of the passivation film.

*3.4. Tribocorrosion Resistance*

Figure 7 indicates the change in the friction coefficient of the coatings and the substrate with sliding time in neutral environment (3.5 wt.% NaCl solution, pH = 7). Two stages were clearly observed during the friction. The first stage is called the initial wear stage, in which the friction coefficient increased dramatically. When the counterpart YG6 started to come into contact with the coating, the contact mainly occurred among the protrusions between the two and greatly increased the motion resistance of the friction pair, causing an increase in the friction coefficient with the sliding time. With the protrusions peeled off from the surface, the contacting area gradually increased, resulting in the wear entering into the stable stage, in which the friction coefficient fluctuates in a small range, showing relative stability. In the stable wear stage, the friction coefficient of the coating without Ta clearly fluctuated more violently than the other two, and the average value was also the highest (coating without Ta: 0.715, coating with Ta: 0.645, and substrate: 0.613). The wear profiles of the substrate and the coatings are indicated in Figure 8, and the corresponding wear volumes and wear rates are shown in Table 5. The wear rates of the coatings were clearly lower than that of the substrate, reduced by 14.8% and 64.5%. As far as the coatings are concerned, the introduction of Ta resulted in the wear rate of the coating being reduced by 58.3%, and the wear rate for the coating without Ta is $4.7458 \times 10^{-4}$ mm$^3$·N$^{-1}$·m$^{-1}$. Feng et al. [20] prepared Ti-Al-(C, N) composite coatings on the surface of TC4 by laser cladding and investigated the tribocorrosion of the coatings in artificial seawater. When the load was 20 N, similar to the load in the present paper, the wear rate was calculated to be $1.04 \times 10^{-3}$ mm$^3$·N$^{-1}$·m$^{-1}$. The coating fabricated in the present paper obviously showed a better tribocorrosion resistance property.

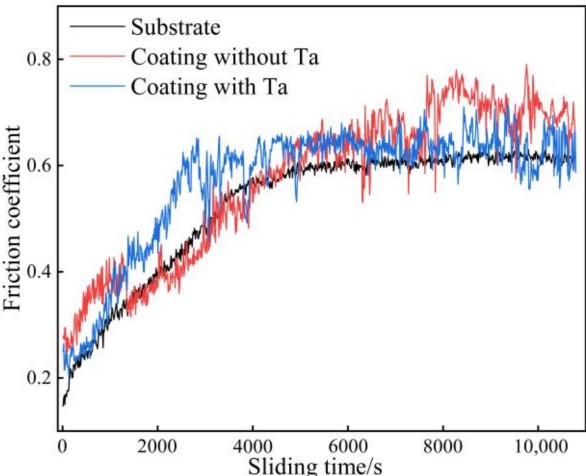

**Figure 7.** The relationship between friction coefficient and sliding time of the substrate and the coatings in neutral environment.

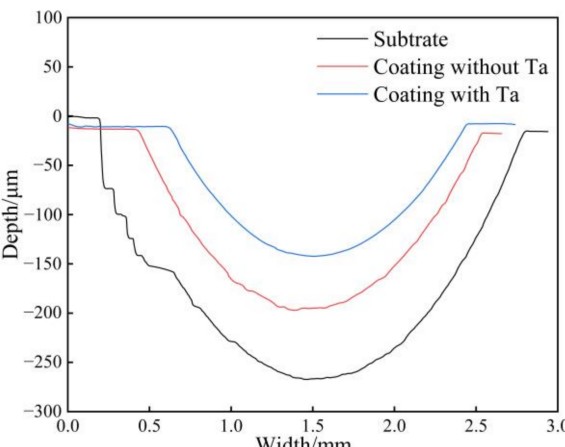

**Figure 8.** Worn profiles of the substrate and the coatings subject to friction in the neutral environment.

**Table 5.** Wear volumes and wear rates of the substrate and the coatings subject to friction in neutral environment.

| Sample | Wear Volume/mm$^3$ | Wear Rate/mm$^3$·N$^{-1}$·m$^{-1}$ |
|---|---|---|
| Substrate | 17.3322 | $1.3374 \times 10^{-3}$ |
| Coating without Ta | 9.6749 | $7.4562 \times 10^{-4}$ |
| Coating with Ta | 6.1505 | $4.7458 \times 10^{-4}$ |

Corrosion could increase the roughness of the material surface by destroying the integrity of grain boundaries or the other microstructure, resulting in the surface nring being easily removed by the counterpart (YG6). Moreover, corrosion could lead to the appearance of pitting pits on the surface of the sample. The stress concentration at the tips of pits could promote the generation and propagation of the cracks, accelerating the loss of coating. According to the conclusion in part 3.3, the coating without and with Ta exhibited more excellent corrosion resistance than the substrate in neutral environment, which could effectively reduce the coating loss caused by corrosion during the sliding process. Meanwhile, the hardness of the coatings was higher than that of the substrate (microhardness: 330 HV$_{0.2}$ for the substrate, 593.28 HV$_{0.2}$ for the coating without Ta, and 617.3 HV$_{0.2}$ for the coating with Ta), which could effectively restrain the plastic deformation, making it hard for the counterpart YG6 to be impressed into the coating surface. The improvement in resistance to corrosion and micro-cutting is responsible for the decrease in the wear rate of the coatings (especially the coating with Ta).

Figure 9 indicates the wear morphologies of the substrate and coatings subjected to wearing in 3.5 wt.% NaCl solution. The chemical compositions of the marked zones in Figure 9 are listed in Table 6. As shown in Figure 9a, the wear surface of the substrate was severely rough and full of fine grooves that were parallel to the sliding direction of the counterpart (YG6). The grooves may have originated from micro-cutting or plastic deformation. A large amount of irregular wear debris in different sizes were attached to the substrate surface due to the white boundaries between the two. EDS was applied to analyze the two zones (substrate in Zone 1 and debris in Zone 2), and the results are demonstrated in Figure 9b. Zone 1 was mainly composed of Ti (73.79 at.%), Al (7.74 at.%), and V (4.49 at.%). Moreover, approximately 4.02 at.% of O was detected, indicating that the wear surface suffered from oxidization during sliding. The elements in Zone 2 were completely similar to those in Zone 1, and no elements of YG6 were detected, indicating that the wear debris was mainly derived from the substrate. However, the O content of Zone 2 (26.57 at.%) was much higher than that of Zone 1, indicating that the exfoliated debris underwent more serious oxidation. Therefore, the wear mechanism of the substrate could be determined as a combination of serious micro-cutting, active dissolution, and

oxidation. The wear morphology of the coating without Ta was smooth, and the width and depth of the grooves were reduced compared with those of the substrate (Figure 9c,d). The number in wear debris was greatly reduced compared with that on the substrate surface. The high-magnification BSE image revealed that the surface of the wear debris was considerably smooth. Some fine cracks were also observed in the debris. The EDS results also demonstrated that the wear debris originated from the coating due to the same elements in Zones 3 and 4 and suffered from serious oxidization. Owing to the high hardness of the coating, the debris from micro-cutting was broken into fine particles, and the coating provided strong support to allow them to easily roll and conglomerate into a large sheet-like debris with a smooth surface during wearing. The sheet-like debris could shield the coating from being destroyed. However, the generated alternating stress during sliding caused the initiation and propagation of cracks. Moreover, a large number of corrosion pits were formed in the wear tracks, demonstrating that a portion of loss came from the active dissolution. As an abnormal phenomenon, the corrosion pits were not observed on the wear surface of the substrate with lower corrosion resistance than the coating, which should be attributed to the serious plastic deformation of the substrate surface. After Ta was introduced into the coating, only extremely fine scratches could be observed on the surface (Figure 9e,f), and the corrosion pits on wear tracks were also reduced, which resulted from the further improvement in hardness and corrosion resistance. Moreover, the sheet-like debris also decreased in number due to a portion of them being crushed into a large number of small dark gray particles uniformly distributed on the wear surface. The wear mechanism of the coatings is similar to that of the substrate. However, serious micro-cutting was transformed into moderate micro-cutting of the coating without Ta, and finally into slight micro-cutting of the coating with Ta.

**Table 6.** Chemical compositions of marked zones shown in Figure 9.

| Sample | Zones | Elements (at.%) | | | | | | | | | |
|---|---|---|---|---|---|---|---|---|---|---|---|
| | | C | O | Al | Ti | V | Cr | Fe | Co | Ni | Ta |
| Substrate | 1 | 10.96 | 4.02 | 7.74 | 73.79 | 4.49 | - | - | - | - | - |
| | 2 | 9.52 | 26.57 | 6.73 | 54.93 | 2.26 | - | - | - | - | - |
| Coating without Ta | 3 | 14.59 | 4.14 | 5.15 | 51.40 | 1.32 | 2.57 | 5.22 | 6.59 | 9.01 | - |
| | 4 | 11.22 | 15.10 | 6.11 | 48.91 | 2.29 | 3.94 | 4.38 | 4.00 | 4.06 | - |
| Coating with Ta | 5 | 20.20 | 4.64 | 6.80 | 43.65 | 2.57 | 3.87 | 2.97 | 2.40 | 1.94 | 10.97 |
| | 6 | 18.02 | 20.47 | 4.89 | 35.96 | 2.05 | 2.95 | 3.32 | 2.91 | 3.37 | 6.06 |

Figure 10 indicates the change in friction coefficient of the coatings and the substrate with sliding time in NaOH solution (pH = 11). The friction coefficient of the substrate maintained a rising trend with the increase in sliding time without the entrance into the stable wear stage, indicating that the substrate remained in the initial wear stage until the end of the test, which may be due to the strong causticity of the solution. The friction coefficient of the coatings fluctuated widely until the sliding time reached 75 min and then entered the relative stable stage. The average friction coefficient of the coatings with Ta was about 0.419, 36.71% less than that of the coating without Ta (about 0.662).

The wear profiles of the substrate and the coatings subjected to NaOH solution (pH = 11) are shown in Figure 11, and the corresponding wear rates are calculated in Table 7. Combining the result of Figure 11 and Table 7, the wear rates of coating without and with Ta were found to be reduced by 59.64% and 61.78% compared with that of the substrate, respectively. This finding indicated that laser cladding endows the substrate with good wear resistance, even in a strong alkaline operating environment, and it could play a good role in substrate protection. Compared with the wear rate of samples subjected to neutral environment, the friction coefficient of the coating with Ta subjected to NaOH solution (pH = 11) decreased by 35.04%, and the wear rate of that increased by 40.77%.

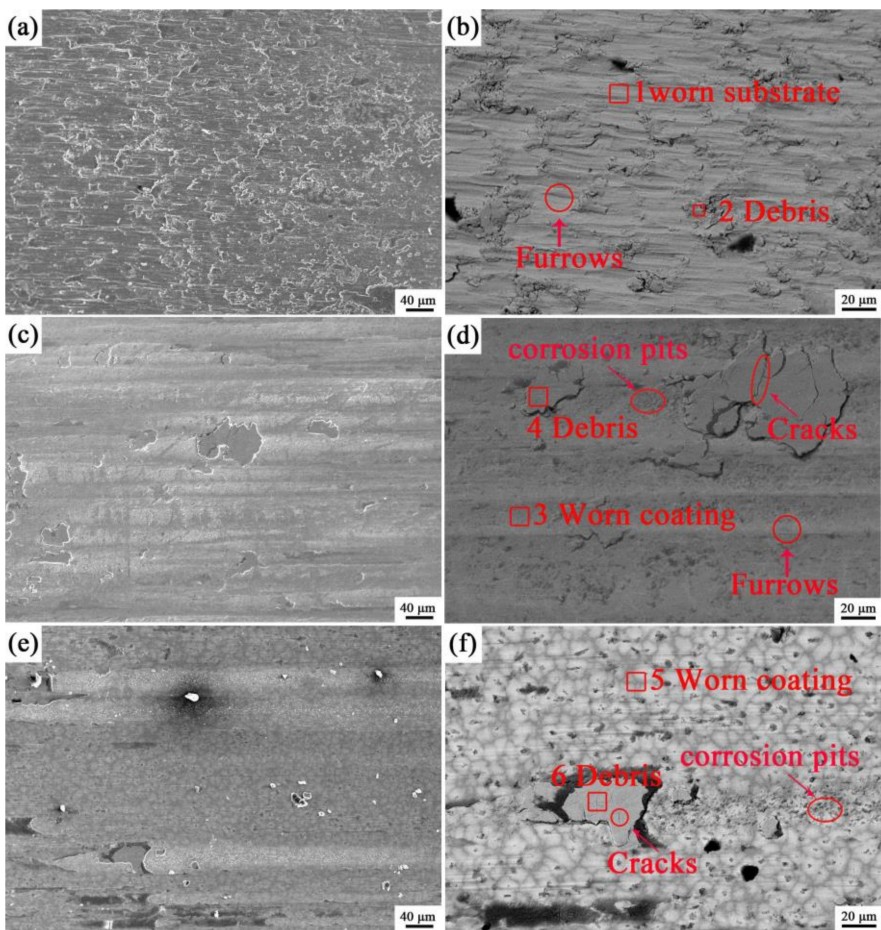

**Figure 9.** Wear morphologies of substrate and coatings subject in neutral environment: (**a**,**b**) substrate; (**c**,**d**) coating without Ta; (**e**,**f**) coating with Ta.

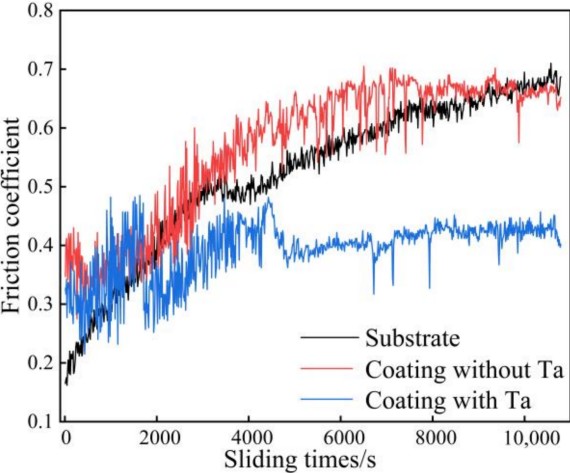

**Figure 10.** The relationship between friction coefficient and sliding time of the substrate and the coatings in alkaline environment.

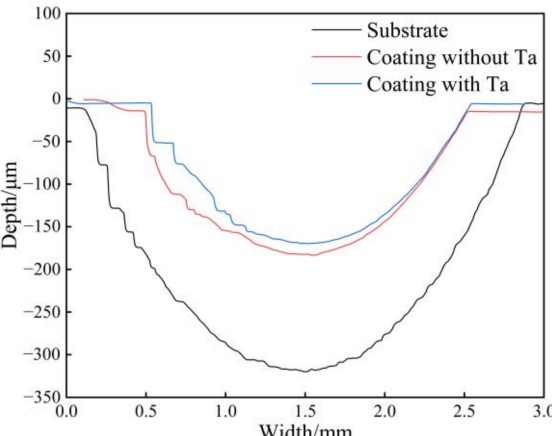

**Figure 11.** Worn profiles of the substrate and the coatings subject to friction in the alkaline environment.

**Table 7.** Wear volumes and wear rates of the substrate and the coatings subject to friction in the alkaline environment.

| Sample | Wear Volume/mm$^3$ | Wear Rate/mm$^3 \cdot$N$^{-1} \cdot$m$^{-1}$ |
|---|---|---|
| Substrate | 22.6562 | $1.7482 \times 10^{-3}$ |
| Coating without Ta | 9.1434 | $7.0551 \times 10^{-4}$ |
| Coating with Ta | 8.6583 | $6.6808 \times 10^{-4}$ |

Mechanical wear and corrosion wear occur at the same time and interact with each other in the friction wear testing in corrosive media. A thin passive film was quickly formed on the surface of coatings when it came into contact with the corrosion solution. When the counterpart contacted the surface of the samples, the local stress sharply increased, leading to an increase in surface roughness. Meanwhile, the passivation film was destroyed, and the surface was re-exposed to the solution, which accelerated the corrosion of the sample surface and made the friction coefficient show a rising trend at the beginning. However, with continuous friction, the debris on the wear surface was brought out of the wear area by the continuous flow of liquid, and the formation and destruction of the passivation film reached a dynamic equilibrium state. Thus, the fluctuation of the friction coefficient gradually decreased. The decrease in the wear rate of the coating without Ta in alkaline environment may be due to the passivation behavior of Ni and Cr in an alkaline solution, whereas the corrosion resistance performance of Ta in strong alkali is poor.

The wear morphologies of the substrate and coatings subjected to NaOH solution (pH = 11) are shown in Figure 12, and Table 8 shows the EDS results for the marked areas in Figure 12. Compared with the wear surface of the coatings, that of the substrate was extremely rough, as obviously shown in Figure 12a,b. Moreover, a large number of grooves were parallel to the sliding direction. The formation mechanism of these grooves is similar to the furrows in Figure 9a,b. The surface protrusions of the hard counterpart contributed to the micro-cutting effect on the substrate, and the coarse and blunt protrusions caused the intense plastic deformation of the substrate, leaving the patterns in the figure. The blocky debris (Zone 2) that left the substrate surface still adhered to the substrate surface under the action of micro-cutting. The elements in Zone 2 were completely the same as those in Zone 1, and the atomic percentages of each element in the two zones were similar. The contents of O in Zones 1 and 2 are 15.86 at.% and 17.26 at.%, respectively, suggesting that the surface of the substrate was oxidized. The wear mechanism of the substrate could be identified as the combination of micro-cutting, oxidation, and active dissolution. The wear surfaces of the coating without Ta was clearly smoother than that of the substrate, and the width of the furrows was thinner (Figure 12c,d). This finding indicated the wear loss of the coating caused by micro-cutting was reduced. A large area of peeling could be observed

on the surface of the coating without Ta, leaving sheet-like debris. Moreover, cracks and pitting pits could be observed, and the grain boundaries of the coating could be faintly seen. The result of EDS in the coating surface (Zone 3) and the debris (Zone 4) revealed that the degree of oxidation in Zone 2 was more serious because the content of O in Zone 4 was twice than that in Zone 3. Compared with the wear morphology of the substrate, the plastic deformation caused by micro-cutting was not obvious because of the increase in coating hardness. After Ta was introduced into the coating, only scratches could be observed on the surface (Figure 12e,f), indicating that the coating was strongly resistant to micro-cutting. More corrosion pits and cracks appeared on the surface of the coating. The cracks, which were formed under the effect of shear stress, were perpendicular to the sliding direction. After the passivation film on the surface was destroyed, the solution entered the cracks to accelerate the crack expansion and increase the loss of the coating. The debris particles that left the coating surface during sliding piled up and adhered to the surface under the action of hard counterpart. According to the EDS result of Zone 6, the main elements of grey block were Ti (43.49 at.%), O (16.29 at.%), and C (18.26 at.%). Compared with that in the coating without Ta, the content of O increased, indicating that more oxidation reactions occurred. However, in the formed oxides, $Ta_2O_5$ was compact, and it showed good wear resistance and corrosion resistance, leading to excellent protection of the surface of the coating. Therefore, the wear mechanism of the coating with Ta could be identified as the combination of slight micro-cutting, oxidation, and active dissolution. Furthermore, Ta plays a positive role in improving the resistance to micro-cutting.

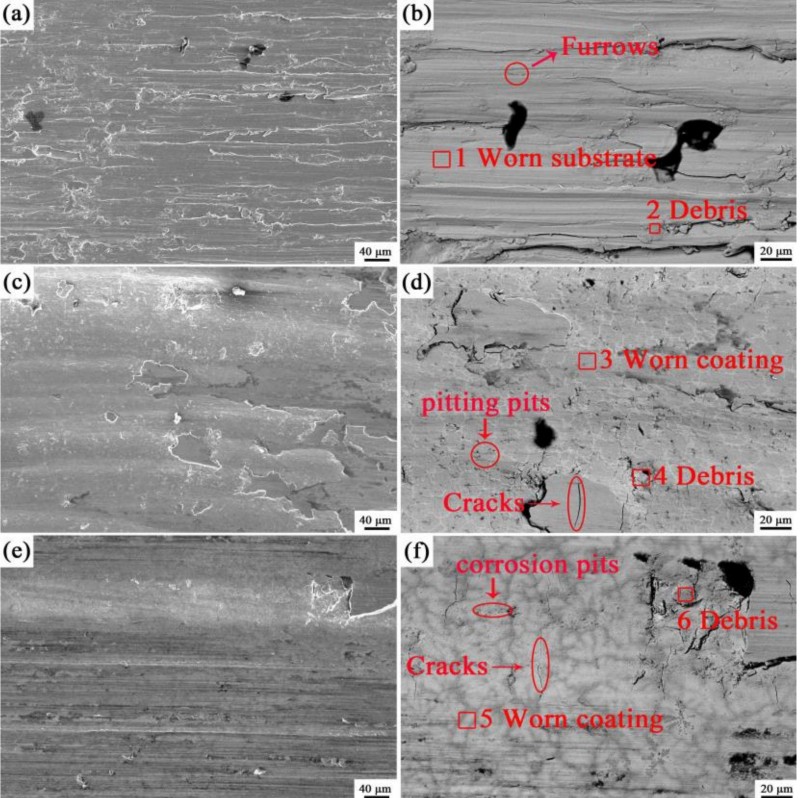

**Figure 12.** Wear morphologies of substrate and coatings subject in NaOH solution (pH = 11): (**a**,**b**) substrate; (**c**,**d**) coating without Ta; (**e**,**f**) coating with Ta.

**Table 8.** Chemical compositions of marked zones shown in Figure 12.

| Sample | Zones | Elements (at.%) | | | | | | | | | |
|---|---|---|---|---|---|---|---|---|---|---|---|
| | | C | O | Al | Ti | V | Cr | Fe | Co | Ni | Ta |
| Substrate | 1 | 10.57 | 15.86 | 8.08 | 62.40 | 3.09 | - | - | - | - | - |
| | 2 | 10.49 | 17.26 | 8.11 | 61.47 | 2.67 | - | - | - | - | - |
| Coating without Ta | 3 | 19.39 | 3.32 | 7.58 | 51.14 | 2.38 | 4.25 | 4.09 | 5.09 | 2.79 | - |
| | 4 | 19.70 | 7.63 | 6.21 | 47.41 | 1.77 | 3.55 | 3.88 | 5.85 | 3.99 | - |
| Coating with Ta | 5 | 18.66 | 8.10 | 6.34 | 47.01 | 1.74 | 2.16 | 3.71 | 3.63 | 4.57 | 4.08 |
| | 6 | 18.26 | 16.29 | 5.56 | 43.49 | 2.11 | 2.11 | 2.60 | 2.47 | 2.70 | 4.4 |

## 4. Conclusions

In this paper, CoCrFeNiTa$_x$ ($x$ = 0, 1) was prepared on the surface of TC4, and the results are as follows:

(1) The phases of the coating were composed of primary $\alpha$(Ti), eutecticum [$\alpha$(Ti) + Ti$_2$Ni], and reinforcement TiC. The addition of Ta resulted in an increased number of TiC and eutecticum.

(2) The addition of Ta obviously alleviated corrosion on the surface in neutral and alkaline environments. The E$_{corr}$ and i$_{corr}$ for the coating with Ta in these environments were as follows: E$_{corr}$: $-0.360$ V, i$_{corr}$: $3.507 \times 10^{-7}$ A·cm$^{-2}$ in a neutral environment; and E$_{corr}$: $-0.468$ V, i$_{corr}$: $7.70 \times 10^{-7}$ A·cm$^{-2}$ in an alkaline environment.

(3) Two coatings showed excellent corrosion wear resistance in neutral (3.5 wt.% NaCl solution, pH = 7) and alkaline (NaOH solution, pH = 11) environments. The wear rate showed a clearly downward tendency in the two environments (in neutral environment: $1.3374 \times 10^{-3}$ mm$^3$·N$^{-1}$·m$^{-1}$ for TC4, $7.4562 \times 10^{-4}$ mm$^3$·N$^{-1}$·m$^{-1}$ for coating without Ta, and $4.7458 \times 10^{-4}$ mm$^3$·N$^{-1}$·m$^{-1}$ for coating with Ta; in alkaline environment: $1.7482 \times 10^{-3}$ mm$^3$·N$^{-1}$·m$^{-1}$ for TC4, $7.0551 \times 10^{-4}$ mm$^3$·N$^{-1}$·m$^{-1}$ for coating without Ta, and $6.6808 \times 10^{-4}$ mm$^3$·N$^{-1}$·m$^{-1}$ for coating with Ta). The wear mechanism of the samples is a combination of serious micro-cutting, active dissolution, and oxidation, and the introduction of Ta effectively improved the resistance to micro-cutting.

**Author Contributions:** Formal analysis, Y.Y.; Investigation, Y.Y.; Resources, J.L. (Jun Li); Writing—original draft, Y.Y.; Supervision, R.L., M.S. and J.L. (Jing Li); Project administration, J.L. (Jun Li). All authors have read and agreed to the published version of the manuscript.

**Funding:** This research was funded by the Natural Science Foundation of Shanghai, China (20ZR1422200) and Class III Peak Discipline of Shanghai—Materials Science and Engineering (High-Energy Beam Intelligent Processing and Green Manufacturing).

**Institutional Review Board Statement:** Not applicable.

**Informed Consent Statement:** Not applicable.

**Data Availability Statement:** Not applicable.

**Conflicts of Interest:** The authors declare no conflict of interest.

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
