# Peer review of "Investigation into Corrosive Wear of the CoCrFeNiTax Laser-Clad Coatings on TC4 in the Neutral and Alkaline Circumstance"

_coatings, doi:10.3390/coatings13010105_

Round 1

Reviewer 1 Report

1- The article requires extensive English editing and proofreading.

2- What is TC4 and why are these properties important for its application?

3- How did your study differ from the existing literature?

4- In Figure 2, label the phases, coatings, and substrates.

5- Data in Table 1 that show elements for points in Figure 3, would be more straightforward if the data is analyzed and labeled the phases or compounds in Figure 3.

6- Any measurement or analysis of the coating thickness?

7- The same comment for Table 5 to Figure 9, would be more straightforward if the data is analyzed and labeled the phases or compounds in Figure 9.

8- How does the wear rate compare to other literature? What is the coating performance?

9- The same comment for Table 7 to Figure 12, would be more straightforward if the data is analyzed and labeled the phases or compounds in Figure 12.

10-In conclusion, remove the first sentence. It is not related to your work. "This section is not mandatory but can be added to the manuscript if the discussion is unusually long or complex.".

Reviewer 2 Report

Review of the manuscript "Investigation into Corrosive Wear of the CoCrFeNiTax Laser-clad coatings on TC4 in the Neutral and Alkaline Circumstance" submitted for publication on "Coatings". In the manuscript the effect of Ta addition on CoCrFeNi coating has been investigated. After that the wear performance in neutral and alkaline conditions has been investigated.

The manuscript appears interesting, well organized and with a deep focus on the microstructure of the system coating + substrate with XRD, potentiodynamic method, SEM, XPS and wear characterization.

I don't have any particular concerns.

The manuscript can be accepted after the following minor revisions:

In the list of authors the author J. Li is indicated twice. Please check!

Table 1 is not called in the main text.

Reviewer 3 Report

What is metallurgical binding? (check line 37 in the introduction)

The introduction is too long. It should be streamlined properly to reflect the justification and research gap from previous research, and this should serve as the main background of the study

During the specimen preparation, why was the substrate subjected to annealing since the study is not focused on it?

The authors have shown that the starting powders were dispersed within one another in a milling machine. What do you have balls in the milling jar since mechanical alloying is not one of the main objectives of the research? 

What is YG6 alloy ball? the surface roughness of this counterface  ball should be included (if available)

The peaks in the XRD pattern should be identified and labelled properly

The cross-sectional of the coatings shown in Fig 2 are almost similar. Since this was examined under SEM, the magnification should be increased to reveal the real structure of the Ta coating

The caption in Fig 3 is confusing. It should be rearranged. Moreover, I think showing either SE or BSE is adequate to show the morphology of the coatings

Fig. 7 does not depict a typical tribocorrosion experiment. The trend of the curve should have three specific regions: Friction start (before load is applied), sliding process (during load application) and friction end (where the applied load is removed). You can check the literature for more information on this

conclusively, the manuscript would be in a better form if the authors are able to incorporate all the recommended changes and subject the manuscript to an English editing service. They also need to correct some of the procedures adopted during the experimentation

Reviewer 4 Report

Overall an interesting publication. The possibility of changing the resistance to corrosion and wear by applying a different material to the surface of the alloy was shown.

My comments:

"When a material is in contact with  another material in a corrosion system, the material with a more negative Ecorr is inclined  to act as the cathode and suffers from serious corrosion by losing electrons" - according to my knowledge, in the system of two metalic ( or non-metalic i.g. graphite) materials with different corrosion potentials, the one with a lower potential is the anode and is subject to corrosion ???  

Is the Pilling–Bedworth ratio (PBR) applicable in aqueous solutions to assess the possible behavior of oxide layers???

In the NaOH solution, it would be necessary to check which of the coating components react with the hydroxide. Because by observing the Pourbaix diagrams, most of the components in alkaline solutions are passivated. Only aluminum and vanadium behave differently. And can the resulting coating be treated as metals covering the surface of the titanium alloy or as a new quality that behaves differently than its individual components???

"This section is not mandatory but can be added to the manuscript if the discussion is unusually long or complex" - I guess you can safely drop it.

Round 2

Reviewer 1 Report

Changes have been made according to the previous comments.

Reviewer 3 Report

Accept